# Dipolar magnetostirring protocol for three-well atomtronic circuits

Héctor Briongos-Merino[1,2*], Felipe Isaule[3],
Montserrat Guilleumas[1,2] and Bruno Juliá-Díaz[1,2]

**1** Departament de Física Quàntica i Astrofísica, Facultat de Física,
Universitat de Barcelona, Martí i Franquès 1, E-08028 Barcelona, Spain
**2** Institut de Ciències del Cosmos, Universitat de Barcelona,
Martí i Franquès 1, E-08028 Barcelona, Spain
**3** Instituto de Física, Pontificia Universidad Católica de Chile,
Avenida Vicuña Mackenna 4860, Santiago, Chile.

★ hbriongos@fqa.ub.edu

## Abstract

We propose a magnetostirring protocol to create persistent currents on an annular system. Under this protocol, polar bosons confined in a three-well ring circuit reach a state with high average circulation. We model the system with an extended Bose-Hubbard Hamiltonian and show that the protocol can create circulation in an atomtronic circuit for a range of tunable parameters. The performance and robustness of this scheme are examined, in particular considering different interaction regimes. We also present a method for predicting the optimal protocol parameters, which improves protocol's scalability and enables its application to systems with large numbers of bosons. This overcomes computational limitations and paves the way for exploring macroscopic quantum phenomena.

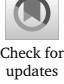

# 1  Introduction

Superfluidity, a hallmark of quantum fluids, is characterized by frictionless flow and exotic phenomena such as quantized vortices and persistent currents [1]. In this line, Bose-Einstein condensates (BECs) provide an ideal platform for exploring these effects due to their macroscopic quantum coherence [2, 3]. In particular, persistent currents, analogous to superconducting loops, can offer insights into the stability and quantum phase coherence in ring-shaped condensates [4].

Recently, the atomtronics field has emerged as a promising platform to study the rich quantum phenomena offered by ultracold atomic gases and to control them for developing quantum devices [5, 6]. This includes persistent currents of neutral matter waves, which convey a technological significance and are intensively studied within atomtronics [7]. Indeed, Superconducting Quantum Interference Devices (SQUIDs) have already been realized experimentally [8, 9]. In particular, atomic SQUIDs are very promising tools in quantum sensing [10] as compact Sagnac interferometers [11, 12] or gyroscopes [13, 14], as well as candidates for the realization of the atomic analog of the Mooij-Harmans qubit [15–17].

The creation of vortices in BECs with different geometries, such as in ring-shaped configurations, can be realized by multiple techniques [18, 19]. These range from potential stirring [20] to suitable Raman transitions [21] and phase imprinting techniques. The latter employ commercially available devices for light sculpting, such as Spatial Light Modulators (SLM) or Digital Mirror Devices (DMD) [22, 23].

A potential additional method for controlling atomtronic circuits is the use of ultracold dipolar particles, as these interact through a tunable dipole-dipole interaction. Dipolar systems can be achieved with either magnetic atoms, such as dysprosium [24, 25], erbium [26], chromium [27], and europium [28], or with polar molecules [29–31]. The dipolar interaction is long-range and anisotropic, with head-to-tail dipoles attracting each other and side-by-side dipoles repelled. These properties give rise to very rich phenomena, even for the smallest systems [32–34].

In this work, we propose a protocol for creating persistent currents as an alternative to the actual state-of-the-art methods. It is based on the magnetostirring technique, which was developed for generating vortices in dipolar gases [35–37]. This technique exploits the anisotropic long-range interactions to induce asymmetry in the system, called magnetostriction [38], to later rotate the direction of polarization, which induces a rotation in the condensate. This method was employed to produce the first observed vortices in a dipolar BEC [35] and has been further used to explore vortex lattices in the dominantly dipolar regime for dipolar BECs and supersolids [39, 40]. Rapid magnetostirring is also known to be able to produce systems where atoms remain stationary, but the dipole moments rotate rapidly [41]. This leads to a time-averaged dipole-dipole interaction, known as an anti-dipolar interaction, where head-to-tail anti-dipoles repel and side-by-side anti-dipoles attract.

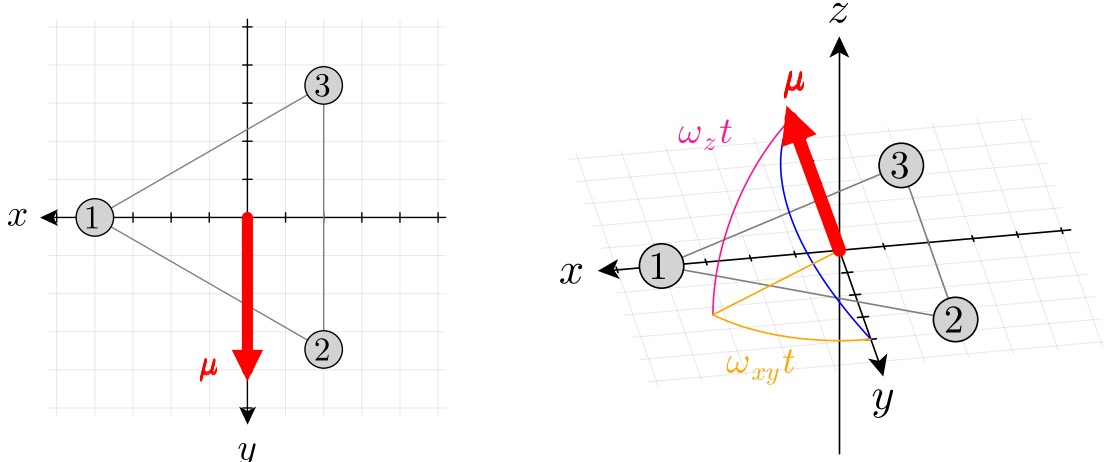

Figure 1: Schematic representation of the triple well system (in the $x$-$y$ plane, left panel) and the movement of the dipole orientation $\boldsymbol{\mu}$ during the protocol (right panel).

We consider a fully connected triple-well ring, the smallest system that supports angular momentum with a superfluid phase [42, 43]. The ring is filled with dipolar bosons to examine the creation of circulation with the magnetostirring-based protocol. We model the system with an extended Bose-Hubbard Hamiltonian for three sites in triangular geometry, to then solve the time evolution of the system and study the creation of a persistent circulation. We find the optimal physical parameters for inducing circulation and study the scheme's robustness, as well as its feasibility in systems with many bosons.

This work is organized as follows: Section 2 introduces the system and the extended Bose-Hubbard Hamiltonian. Section 3 presents the proposed protocol to create circular currents. Then, in Section 4, we discuss the results obtained, analyzing the evolution of the system and the performance of the protocol. Finally, Section 5 presents the conclusions and outlook of this work.

## 2 Theoretical framework

We consider $N$ dipolar bosons confined in a ring formed by 3 sites, as depicted in Fig. 1. We model the system with the following extended Bose-Hubbard Hamiltonian [32, 33, 44, 45]

$$\hat{H} = -J \sum_{j=1}^{3} \left( \hat{a}_{j+1}^{\dagger} \hat{a}_j + \hat{a}_j^{\dagger} \hat{a}_{j+1} \right) + \frac{U}{2} \sum_{j=1}^{3} \hat{n}_j \left( \hat{n}_j - 1 \right) + \sum_{j=1}^{3} \sum_{k \neq j}^{3} \frac{V_{jk}}{2} \hat{n}_j \hat{n}_k, \tag{1}$$

where $\hat{a}_j (\hat{a}_j^{\dagger})$ are the bosonic annihilation (creation) operators for the $j$-th site, $\hat{n}_j = \hat{a}_j^{\dagger} \hat{a}_j$ is the particle number operator, $J$ is the nearest neighbors tunneling strength, $U$ is the on-site interaction strength, and $V_{jk}$ is the strength of the dipolar interaction between sites $j$ and $k$. The dipolar interaction term is anisotropic and depends on the relative orientation of the dipoles. In this work, we consider that all individual dipoles are aligned in the same direction at any given time, which is achieved by polarizing them with an external electromagnetic field.

Under this assumption, the dipolar interaction has the form

$$V_{jk} = \frac{U_d}{|\mathbf{r}_j - \mathbf{r}_k|^3} \left\{ 1 - 3 \left[ \frac{\boldsymbol{\mu} \cdot (\mathbf{r}_j - \mathbf{r}_k)}{|\mathbf{r}_j - \mathbf{r}_k|} \right]^2 \right\}, \tag{2}$$

where $U_d$ is the dipolar coupling constant, $\boldsymbol{\mu}$ is the unitary dipole orientation, and $\mathbf{r}_j - \mathbf{r}_k$ is the vector that points from the $k$-th to the $j$-th site. Due to the geometry of the fully connected three-well system, the term $|\mathbf{r}_j - \mathbf{r}_k|^3$ is constant for all pairs of sites and can be absorbed into $U_d$. Therefore, in the following, we will not explicitly write such dependence, as it will be included in the constant $U_d$.

The Hamiltonian (1) commutes with the number operator $\hat{N} = \sum_{j=1}^3 \hat{n}_j$, i.e. the system is number conserving, and thus we can subtract a factor $(U/2)\hat{N}(\hat{N}-1)$ to the Hamiltonian when working in sectors of well defined $N$. This enables us to rewrite it in the following convenient form [46, 47]

$$\hat{H}' = \hat{H}_J + \hat{H}'_I = -J \sum_{j=1}^3 \left( \hat{a}_{j+1}^\dagger \hat{a}_j + \hat{a}_j^\dagger \hat{a}_{j+1} \right) + \sum_{j=1}^3 \sum_{k \neq j}^3 \left( \frac{V_{jk} - U}{2} \right) \hat{n}_j \hat{n}_k. \tag{3}$$

This transformation explicitly shows the competition between the inter-site (dipolar) and on-site local interaction terms. This competition is highlighted when the dipole orientation $\boldsymbol{\mu}$ is aligned perpendicular to the plane in which the triple well is contained, $\boldsymbol{\mu} = \mu\, \mathbf{e}_z$, with $\mathbf{e}_z$ the normalized vector along the $z$ direction. In this configuration, $\boldsymbol{\mu} \cdot (\mathbf{r}_j - \mathbf{r}_k) = 0 \ \forall j, k$, which simplifies the interaction term of Hamiltonian (3) to the following expression:

$$\hat{H}'_I \big|_{\boldsymbol{\mu} = \mu\, \mathbf{e}_z} = \frac{1}{2} \sum_{j=1}^3 \sum_{k \neq j}^3 (U_d - U) \hat{n}_j \hat{n}_k. \tag{4}$$

Tuning the on-site interaction strength $U$ via Feshbach resonances [48] to match $U_d$ can nullify the interaction term in Eq. (4). This results in a non-interacting Hamiltonian, and thus the system is described by free particles in a three-well ring potential.

To study the onset of persistent currents, we examine the behavior of the current operator. For a real tunneling parameter $J$, the total azimuthal current operator [7, 49], also called the azimuthal circulation operator, is defined in a triple-well circuit as

$$\hat{L}_z = i \frac{2\pi}{3} \frac{JmR^2}{\hbar} \sum_{j=1}^3 \left( \hat{a}_{j+1}^\dagger \hat{a}_j - \hat{a}_j^\dagger \hat{a}_{j+1} \right),$$

where $m$ is the atomic mass of the bosons and $R$ is the radius of the atomtronic circuit. This operator commutes with the tunneling term of the Hamiltonian, $\hat{H}_J$, sharing a common eigenbasis. The eigenvectors and eigenvalues of the $N$-boson system can be constructed as tensor products or sums, respectively, of the single-particle eigenstates and energies. These single-particle eigenstates and energies are given by (see for example [50])

$$\lambda_J(k) = -2J \cos\left( \frac{2\pi k}{3} \right),$$

$$\lambda_L(k) = \frac{4\pi}{3} \frac{JmR^2}{\hbar} \sin\left( \frac{2\pi k}{3} \right),$$

$$\nu_k = \frac{1}{\sqrt{3}} \left( 1, w^k, w^{2k} \right)^T,$$

where $w = e^{2\pi i/3}$ and $k = 0, 1, 2$. The eigenvalues $\lambda_J(k)$ and $\lambda_L(k)$ correspond to the tunneling term of the hamiltonian, $\hat{H}_J$ and the circulation operator, $\hat{L}_z$, respectively. The eigenstates are

Bloch waves written in the single-particle Fock basis $\{|1,0,0\rangle, |0,1,0\rangle, |0,0,1\rangle\}$; where the Fock vectors are noted as, $|n_1, n_2, n_3\rangle$.

At zero temperature and in the absence of interactions, all the bosons populate the same single-particle state. However, interactions or excitations can remove particles from such a single-particle ground state, promoting them into excited ones. This depletes a fraction of the condensate, or may even cause its fragmentation for sufficiently strong interactions. The condensed fraction $f_c$ of the system described by the many-body wave function $\Psi$ is given by the largest eigenvalue of the one-body density matrix operator (OBDM) [51]

$$\hat{\rho}_{j,k} = \frac{1}{N}\langle\Psi|\hat{a}_j^\dagger \hat{a}_k|\Psi\rangle\,, \tag{5}$$

while its associated eigenvector is the so-called natural orbit of the system, the most populated eigenstate. For a singly condensed system, the largest eigenvalue is close to unity ($f_c \simeq 1$), while the others are roughly zero. On the other hand, if the system is fragmented, two or more eigenvalues will have a comparable magnitude [52].

Because the extended Bose-Hubbard model is non-integrable for arbitrary dipolar angles, performing time evolutions is only feasible through perturbative approximations. However, these approximations break down in the strongly interacting regimes explored in this work. Therefore, we need to rely on numerical calculations. In our numerical approach, we construct a Fock basis for $N$ bosons distributed over three sites and apply exact diagonalization techniques. Time evolution is carried out using the fourth-order Runge-Kutta method. The size of this basis grows exponentially with the number of bosons, which limits the systems that are computationally affordable to a few tens of particles [53]. For the system with three sites, we are able to study up to 40 particles within reasonable computing times.

## 3 Circulation creation protocol

The proposed protocol for the creation of circulation in the three-well system consists of a dynamic change of the dipole alignment $\boldsymbol{\mu}$, which breaks the reflection symmetry of the system. Initially, at $t = 0$, the dipoles are polarized in the $y$-axis direction [see Fig. 1]. The initial state is chosen as the ground state of such a configuration, in which sites 2 and 3 concentrate most bosons due to the dipolar attraction [33, 34]. Then, the polarization direction changes over time with a spherical spiral motion parametrized by

$$\boldsymbol{\mu}\cdot\mathbf{e}_x = \sin\left(\omega_{xy}\,t\right)\cos\left(\omega_z\,t\right)\,,$$
$$\boldsymbol{\mu}\cdot\mathbf{e}_y = \cos\left(\omega_{xy}\,t\right)\cos\left(\omega_z\,t\right)\,,$$
$$\boldsymbol{\mu}\cdot\mathbf{e}_z = \sin\left(\omega_z\,t\right)\,,$$

where $\omega_{xy}$ and $\omega_z$ are the free protocol parameters that define the spiral. The parameter $\omega_{xy}$ is the in-plane rotation frequency that controls the velocity of the horizontal movement, while $\omega_z$ controls the vertical movement towards the $z$-direction. This time dependence only affects the dipolar interaction term [equation (2)]. The dynamic protocol finishes when the dipole is aligned with the $z$-axis at a time $t_f = \pi/(2\omega_z)$. After that, the polarization remains fixed in the $z$-direction, that is, perpendicular to the circuit's plane [see Fig. (1)].

The hopping, on-site interaction, and dipolar interaction strengths are constant throughout the protocol. Also, the on-site interaction strength is chosen such that $U = U_d$, thus at the end of the protocol, the system evolves under a free-particle Hamiltonian due to the vanishing of the interaction term in equation (4).

This protocol relies on two main ideas. Firstly, the circulation will be generated by the exchange of bosons between the occupied and empty sites when the dipole orientation starts

to rotate, as the pair of favored wells will change during the in-plane rotation. Secondly, the azimuthal circulation generated is preserved after the end of the protocol as the final configuration maximizes the symmetries of the system and cancels the effects of the interactions.

It should be stated that the protocol parameters $\omega_{xy}$ and $\omega_z$ can be tuned to regulate the system's state at $t_f$. In this work, we focus on the creation of the maximum circulation in the most predictable way, and thus do not analyze other potentially interesting final states that might arise for other values of the parameters.

Throughout this paper, the energy is provided in units of the hopping parameter $J$ and the time in units of $\hbar/J$. The circulation is normalized to the maximum possible circulation of a free boson in a three-site ring, namely $L_0 = \frac{2\pi}{\sqrt{3}} \frac{J m R^2}{\hbar}$.

## 4  Numerical results

### 4.1  Evolution of the system under the protocol

We begin by examining the evolution of some characteristic quantities over time to understand how the system evolves under the protocol and how circulation is produced.

In Fig. 2 we study the evolution of the quantum state $|\Psi(t)\rangle$ under the magnetostirring protocol. Panel a) shows the projection of the instantaneous state of the system onto the eigenstates of the time-dependent Hamiltonian as a function of time. The spectrum of this many-body system is a set of values that change with the dipole orientation, modifying gaps and creating crossings between the instantaneous eigenstates. An important aspect to note is that at the end of the protocol, the energy levels are equidistant, as it is the spectrum of a free Hamiltonian, discussed previously. As required by the protocol, the initial state corresponds to the ground state of the Hamiltonian at $t = 0$. Since the initial polarization lies in the $x - y$ plane and it is aligned along the $y$ axis, the ground state presents an equal population in sites 2 and 3 due to the attractive dipolar interaction, whereas site 1 is empty, as it is shown in panel c).

As the Hamiltonian starts changing, the initial gap closes, allowing the system's state to start overlapping with the excited states. This closing happens because the dipole orientation approaches a configuration for which the ground state of this system is degenerate. In fact, this happens for an in-plane polarization direction perpendicular to one side of the triangle [33,34]. When the state of the system shows a large overlap with excitations, it populates many highly excited states while depopulating the low-lying ones. At the end of the protocol, the state of the system is a combination of highly excited Hamiltonian eigenstates, far from the ground state of the system. Furthermore, the final state is a superposition of multiple circulation states.

The initial value of the azimuthal circulation is, as expected, zero. This value starts increasing around half-time of the protocol when the system's state is no longer the instantaneous ground state. The growth stops before the end of the protocol without oscillations, as the system's Hamiltonian at the end of the protocol preserves the circulation. This change in the expected value of the azimuthal circulation during the protocol is shown in Fig. 2 b). Therefore, the protocol successfully creates a persistent circulation for $t > t_f$.

At the beginning of the protocol, due to the anisotropic character of the dipolar interaction, only two sites are populated (sites 2 and 3). However, as the system evolves with time, the occupation changes, decreasing the population in sites 2 and 3, while increasing it in the initially unpopulated one (site 1). This population evolution is shown in Fig. 2 c). It must be noted that at the end of the protocol, the populations in the three sites are not equal, which is what one would expect for a pure circulation state. Therefore, one can consider this imbalance as a signature of the superposition of circulation eigenstates in the system.

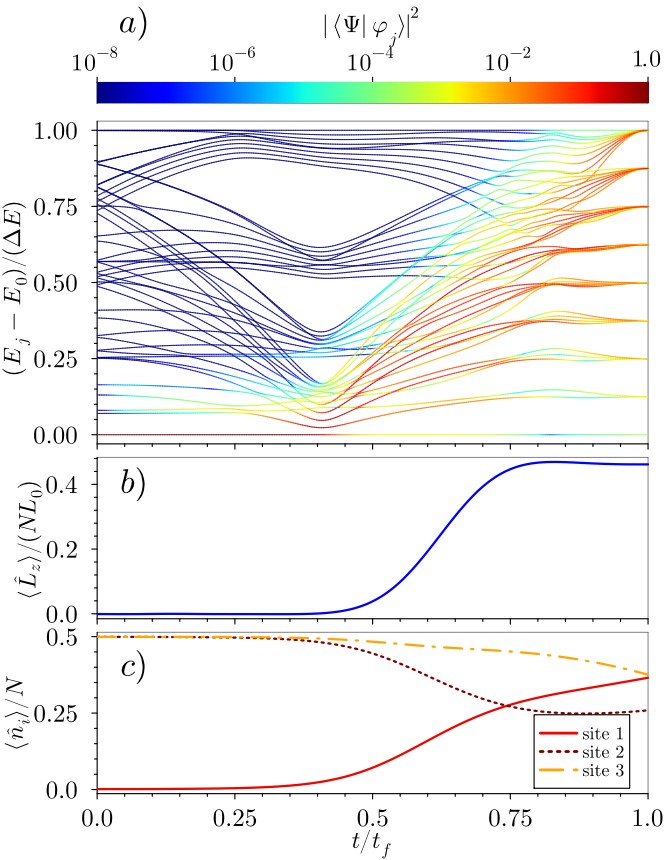

Figure 2: Evolution of the state of a system with $N = 8$ bosons and interaction strengths $U/J = U_d/J = 5.0$ under the magnetostirring protocol with $\omega_{xy} = 1.07\,\hbar/J$ and $\omega_z = 1.32\,\hbar/J$. a) Eigenenergies $E_j$ and projection of the system's state $|\Psi(t)\rangle$ onto the instantaneous eigenstates of the Hamiltonian, $\{|\varphi_j\rangle\}\big|_t$, as a function of time. The color map (upper bar) corresponds to the probability density $|\langle\Psi|\varphi_j\rangle|^2$ in logarithmic scale. The eigenenergies are rescaled on each step to improve visibility. b) Expected value of the azimuthal circulation as a function of time. c) Expected occupation of each of the three sites as a function of time.

## 4.2 Parameter optimization

It is necessary to choose appropriate values for the frequencies $\omega_{xy}$ and $\omega_z$, the two parameters of the time-dependent Hamiltonian, to maximize the circulation produced after the end of the protocol. To explore the parameter landscape, we have computed the expected final azimuthal circulation of the system for a fixed number of bosons and interaction strengths as a function of $\omega_{xy}$ and $\omega_z$. One representative exploration is shown in Fig. 3.

Within the parameter landscape, distinct regimes can be identified. At one extreme, a very fast quench is not expected to induce circulation. This sets an upper limit on the dimensionless parameter $\omega_z\hbar/J$, corresponding to a lower bound on the protocol duration $t_f$. As shown in the figure, when $\omega_z\hbar/J > 10$, the protocol duration $t_f < \frac{\pi\hbar}{20J}$ is too short for the bosons to move into the initially unoccupied well, and no circulation is generated. Conversely, for small values of the azimuthal parameter ($\omega_z\hbar/J < 0.5$), the protocol is sufficiently slow to allow the system to undergo multiple oscillations during the evolution. In this region, the chosen grid is too coarse to resolve the system's continuous evolution, resulting in a noisy-like pat-

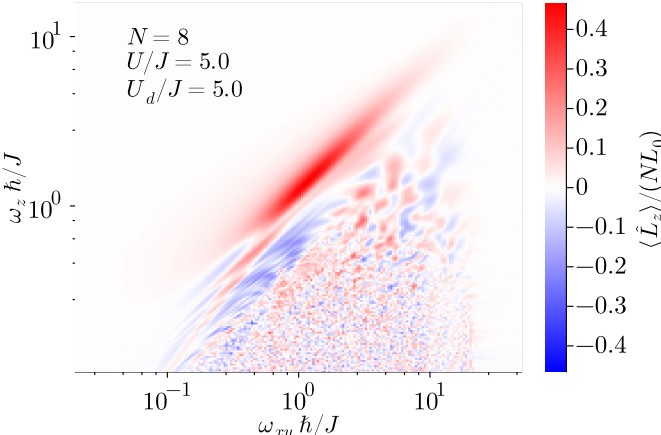

Figure 3: Expected value of the azimuthal circulation at the end of the protocol of a system with $N = 8$ bosons and interactions $U/J = U_d/J = 5.0$ for different combinations of $\omega_{xy}$ and $\omega_z$.

tern. Regarding the parameter $\omega_{xy}$, we observe that for values $\omega_{xy}\hbar/J < 0.1$, no circulation is generated. In this regime, the slow in-plane rotation leads to an adiabatic evolution of the system, allowing it to remain in the instantaneous ground state throughout the protocol. The rotation is not fast enough to induce transitions to excited states. For large rotation frequencies, values $\omega_{xy}\hbar/J > 11$, no circulation is produced. In this regime, the bosons are unable to follow the rapid changes in the polarization direction. As a result, the system's evolution effectively resembles that under a static antidipolar interaction, which is essentially the time-averaged interaction experienced by dipolar gases at high rotation frequencies [41]. Hence, the protocol is effective only within an intermediate frequency range, corresponding to the central region of the figure, where a large lobe of positive circulation (red region) emerges.

We therefore conclude that the protocol exhibits a well-defined parameter range in which circulation is consistently produced. This region, or lobe, is bounded by the upper and lower limits discussed previously. However, its precise location within the parameter space is not fixed, as it depends on both the number of particles and the interaction strength. Consequently, the optimal parameters for inducing circulation vary with the number of bosons and interaction strength. Interestingly, as shown in Appendix A, the values of these optimal parameters can be predicted for large systems.

## 4.3 Circulation's dependence on the number of bosons

As shown previously in Figures 2 and 3, the maximum circulation achieved for a system of eight particles is less than half of the theoretical maximum circulation such a system can support. However, this behavior changes as the number of particles increases. Specifically, the circulation generated by the protocol grows faster than the system's maximum possible circulation, given by $NL_0 = \frac{2\pi}{\sqrt{3}}N\frac{JmR^2}{\hbar}$. This indicates that the protocol becomes increasingly efficient at generating circulation as the number of bosons increases. In Fig. 4 we show the growth of the maximum circulation generated as a function of the number of particles for different interaction strengths with $U = U_d$.

We stress that the points shown in Fig. 4 are obtained for the optimal parameters ($\omega_z$ and $\omega_{xy}$) for the chosen interaction strengths and the number of particles [the lobes discussed in the previous subsection]. The series of points for each interaction strength starts from a different number of bosons because we do not show results for which the optimal parameters are outside the lobes. Finding higher circulation values for a combination of frequencies outside

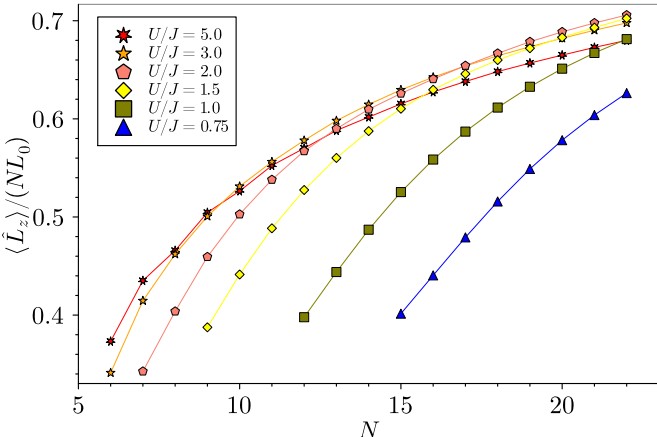

Figure 4: Maximum circulation generated in the system after the protocol for different interaction strengths (indicated in the labels) as a function of the number of bosons $N$. The lines are a guide for the eye.

the lobe only happens for smaller numbers of particles, where the protocol is more difficult to optimize. This further supports the idea that the effectiveness and reliability of the protocol improve as the number of bosons increases.

We also note that the circulation produced for a small number of particles $N < 10$ is usually larger for stronger interactions. However, the increase of the maximum circulation as a function of $N$ is slower with larger $U$. This means that, in systems with a large number of particles, greater circulation is achieved when the interaction parameters are comparable to the hopping strength.

### 4.4 Condensed fraction at the end of the protocol

After characterizing the creation of circulation with different parameters, we now study its connection with the condensed fraction $f_c$ at the end of the protocol, which is extracted from the OBDM [equation (5)]. Naively, one would expect a low condensed fraction at the end of the protocol because of the excitation of the system. One would also expect a decrease of $f_c$

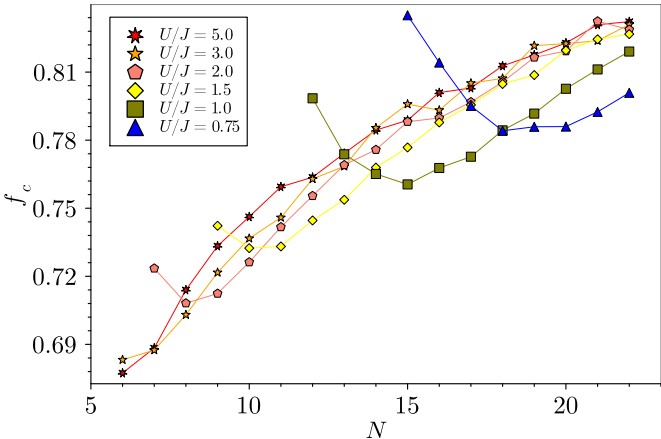

Figure 5: Condensed fraction $f_c$ at the end of the protocol for different interaction strengths (indicated in the labels) as a function of the number of bosons. The lines are a guide for the eye.

for increasing values of the inter-particle interactions. However, we find large values of the condensed fraction in the protocol. Furthermore, we find that $f_c$ grows with the number of particles almost independently of the interactions. This behavior happens due to the cancellation of the on-site and dipolar interactions at the end of the protocol, therefore reducing the fragmentation. This is shown in Fig. 5, where the value of $f_c$ is depicted at the end of the protocol as a function of $N$ for different interaction strengths at the optimal protocol parameters used in Fig. 4.

For each interaction's value, we also observe a local minimum of $f_c$, after which it starts to grow. This is in contrast with the expected average tendency of the condensed fraction. The minimum of $f_c$ is inversely proportional to the value of the interactions, thus recovering the idea that increasing interactions deplete a larger fraction of the system. We include a more detailed analysis of the fragmentation of the system in Appendix B, and a discussion about the validity of describing the system using a condensed mean-field framework.

## 4.5 Robustness of the protocol

In the previous sections, we have followed all the proposed requirements of the protocol. In the following, we analyze the robustness of the protocol under slight modifications of the on-site interactions and the number of particles.

First, we consider a variation of the on-site interaction parameters. In general, we expect a decrease in the circulation generated in systems where the on-site interaction $U$ is larger than the dipolar $U_d$, as the stirring mechanism becomes less effective. On the other hand, a weaker on-site interaction would seem preferable. This is true if the deviation is relatively small, while for larger deviations it fails. Additionally, imperfect cancellation of the interactions at the end of the protocol also reduces the final circulation. This results in the restriction for $U$ and $U_d$ to be similar in order to achieve the largest circulation. The final circulation obtained considering a deviation $\Delta U \equiv U - U_d$ in the on-site interaction strength is presented in Fig. 6. We employ the optimal protocol frequencies $\omega_{xy}$ and $\omega_z$ obtained for $U = U_d$. The behavior is qualitatively similar for all the systems analyzed, all of them showing the previously described features. For positive deviations $\Delta U/J > 0$, the circulation produced in the system decreases as the deviation increases. On the other hand, the final circulation grows while $\Delta U/J$ decreases, but for deviations $\Delta U/J < -0.2$, it starts decreasing.

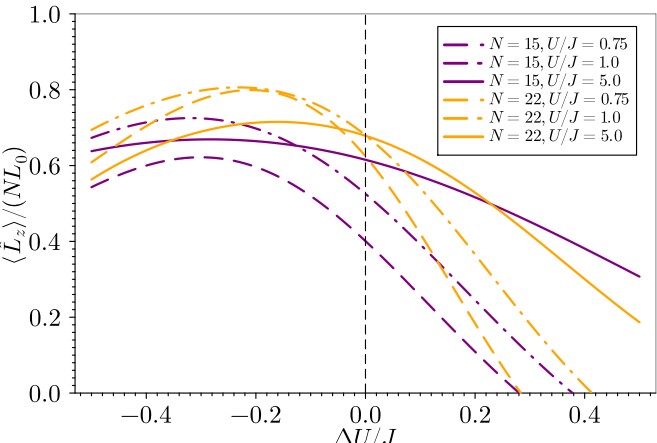

Figure 6: The final value of the circulation as a function of the on-site interaction deviation $\Delta U/J$ for fixed values of $U_d$ and multiple numbers of particles (indicated in the labels), where the protocol parameters are the ones that optimize final circulation at $\Delta U/J = 0$.

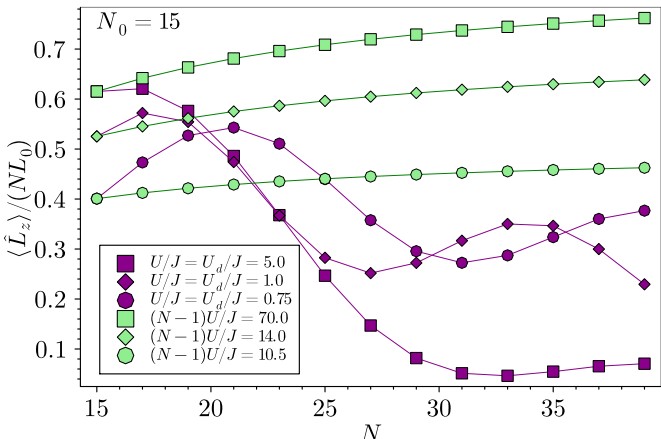

Figure 7: Final value of the circulation as a function of the number of bosons $N$. In purple, the protocol frequencies are fixed for constant $U/J$, while, in green, the protocol frequencies are fixed for constant $(N-1)U/J$. The lines are a guide for the eye.

Having analyzed the robustness of the protocol under a change in the interaction strengths, we now examine its robustness with respect to changes in the number of particles. In Fig. 7 we show the final circulation as a function of $N$, having chosen the optimal parameters for $N_0 = 15$ particles and maintaining $U/J$ (purple markers and lines) and $(N-1)U/J$ (green markers and lines) constant.

For constant interaction strengths $U/J = U_d/J$ (purple markers), the circulation shows a significant decrease when varying $N$. This indicates that, in general, the chosen protocol frequencies do not remain optimal under a change in the number of particles. On the other hand, by maintaining the value of $(N-1)U/J = (N-1)U_d/J$ constant (green markers), the circulation fraction increases with $N$. This means that, to maintain and even increase the circulation for a larger number of particles, it is necessary to maintain a fixed ratio between the interaction strengths $U = U_d$ and $(N-1)/J$. More importantly, this enables one method to predict optimal protocol frequencies for large $N$. Indeed, one can find the optimal frequencies for a small number of particles, and then choose the interaction that maintains $(N-1)U/J$ constant for a larger $N$.

## 5 Conclusions

We have proposed a functional magnetostirring protocol designed to create azimuthal circulation in a dipolar system confined in a triple-well Bose-Hubbard ring. Our method consists in exciting the system via a spherical spiral-like modulation of the polarization direction, which is experimentally accessible by changing the orientation of the polarizing magnetic field. We have shown that this protocol can drive the bosonic system from its ground state to an excited state, achieving an average circulation close to the maximum sustainable by the system. This scheme extends the magnetostirring technique to atomtronic rings, expanding the methods that can be used for the creation of persistent currents in such platforms.

This protocol has only two free parameters, which are the two frequencies that control the dipole orientation's motion. We find that there is a set of parameters where the final circulation is maximized. In addition, both the circulation produced in the system and the final condensed fraction increase with the number of bosons, highlighting that this effect is more efficient in

highly populated systems. We also demonstrate that the protocol remains robust under a small offset in the on-site interaction strengths, making it promising for experimental realization.

Finally, for systems with a large number of particles, we present a practical method to determine the optimal frequencies by studying systems with fewer particles. Specifically, the optimal protocol frequencies can be identified in a small-boson system with the same interaction parameter $(N-1)U/J = (N-1)U_d/J$ as the desired system. These frequencies can then be applied to systems with significantly more particles. This property enhances the scalability of the protocol, as it enables to reach systems computationally intractable to simulate using the Bose-Hubbard model. This opens new avenues for the study of macroscopic quantum phenomena.

These results support our protocol as a feasible alternative to the already existing techniques for generating persistent currents, setting it as a powerful tool for future developments in the field of atomtronics. Future research could extend this work to larger discrete rings, for which we expect the protocol to remain effective. Another promising extension involves applying the protocol to continuous toroidal condensates, in which the protocol parameter $\omega_{xy}$ would be lower-bounded by the critical rotation frequency of the system. In the field of atomtronics, these results also offer new possibilities for the realization of rotation sensors [12, 14].

## Acknowledgments

We acknowledge useful and constructive discussions with Abel Rojo-Francàs.

**Funding information** This work has been funded by Grant PID2023-147475NB-I00 funded by MICIU/AEI/10.13039/501100011033 and FEDER, UE, by Grant No. 2021SGR01095 from Generalitat de Catalunya, and by Project CEX2019-000918-M of ICCUB (Unidad de Excelencia María de Maeztu). H. B.-M. is supported by FPI Grant PRE2022-104397 funded by MICIU/AEI/10.13039/501100011033 and by ESF+. F.I. acknowledges funding from ANID through FONDECYT Postdoctorado No. 3230023.

## A Parameter prediction

As discussed in Section 4.5, it is possible to predict optimal protocol parameters for computationally intractable systems by exploiting the behavior shown in Fig. 7. For fixed values of the protocol frequencies, the circulation introduced in the system grows with the number of particles provided that $(N-1)U/J = (N-1)U_d/J$ remains constant. This means that optimal parameters found for a smaller, computationally inexpensive system with few particles $N_1$ can be reused to induce circulation in a larger system with $N_2$ particles, more complex or impossible to compute, as long as the following condition is met:

$$\frac{N_2-1}{N_1-1} = \frac{U_1/J_1}{U_2/J_2},$$

where $U_1$ ($J_1$) and $U_2$ ($J_2$) are the on-site interaction (hopping) strengths for the first and second system, respectively.

In Fig. 8 we show the optimal frequencies of the protocol for the data presented in Figs. 4 and 5. These frequencies follow a decaying power-law trend as $(N-1)U/J$ increases, which corresponds to a longer final protocol time. The figure also includes a tentative fit to this trend, to facilitate the prediction for high values of the parameter. This can provide a useful reference for estimating optimal parameters in future research and applications.

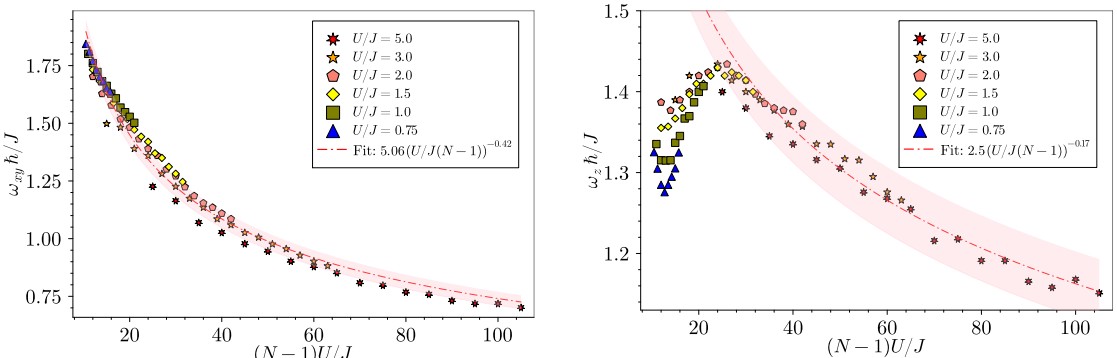

Figure 8: Optimal protocol frequencies $\omega_{xy}$ (left panel) and $\omega_z$ (right panel) as a function of $(N-1)U/J$ for the interaction strengths indicated in the labels. The area shaded in red is the confidence interval of the fitting.

## B  Mean-field description

### B.1  Coherent mean-field model

When the system is fully condensed, the state of $N$ bosons can be expressed as a product of $N$ identical single-particle states. In our three-well system, the single-particle state $\psi$ depends only on three complex coefficients, $\chi_i$, that represent the projection of the single-particle state onto each of the sites of the ring. Thus, the many-body state takes the form:

$$|\Psi\rangle = \bigotimes_{j=1}^{N} |\psi\rangle, \qquad |\psi\rangle = (\chi_1, \chi_2, \chi_3)^T,$$

which evolves under a coherent mean-field set of equations [54, 55]:

$$i\hbar \frac{\partial \chi_j}{\partial t} = -J\left(\chi_{j+1} + \chi_{j-1}\right) + (N-1)\sum_{k \neq j}\left(V_{jk} - U\right)|\chi_k|^2 \chi_j.$$

In this evolution, the condensed fraction remains one throughout the evolution.

We stress that these equations provide only an approximation for our model. This is because time-dependent Hamiltonians and interactions usually deplete a fraction of the condensate, as discussed in the main text. However, previous studies [34] have shown that dipolar interactions prevent fragmentation for comparable on-site interaction strengths. As a result, the mean-field equations could still provide a reliable approximation within certain parameter regimes.

This mean-field model represents the state of the system with only three parameters, and its computational cost is independent of the number of bosons $N$ in the system. Therefore, it provides an efficient approximation for studying systems with a large number of particles.

### B.2  Coherent mean-field results

First, to benchmark the mean-field model, in Fig. 9 we show the circulation at the end of the protocol for the same parameters used in Fig. 3, where results were computed exactly. The overall circulation pattern closely resembles that of the exact results presented in the main text. While the qualitative range of relevant parameters remains similar, the specific pattern of circulation differs. Nevertheless, the circulations obtained in the mean-field model still show a large lobe of positive circulation in the same parameter region, which can help in predicting the optimal protocol frequencies. However, it is important to highlight that in the

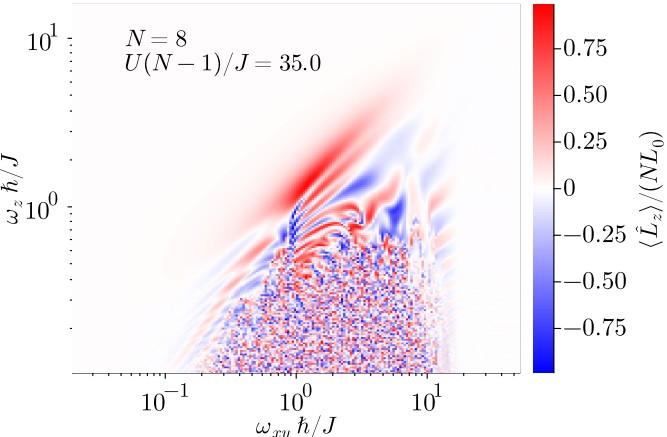

Figure 9: Expected value of the azimuthal circulation at the end of the protocol of a system with $N = 8$ and $U/J = U_d/J = 5$, leading to $(N-1)U/J = (N-1)U_d/J = 35.0$, for multiple combinations of $\omega_{xy}$ and $\omega_z$. Computations are done using the mean-field model.

region where the pattern is not well-resolved, the mean-field model significantly overestimates the circulation compared to the exact Bose-Hubbard calculations.

This discrepancy between the results obtained using the two models indicates that the assumption that the system remains fully condensed does not always hold. There is no guarantee that the system will remain condensed under all protocol frequencies.

## B.3 Validity of the mean-field model

To further test and compare whether and in which region the mean-field model can be used to calculate the results for the stirring protocol, in Fig. 10 we show the final condensate fraction computed using the Bose-Hubbard model. We consider a condensate fraction threshold of 0.75 as a qualitative boundary to distinguish between condensed and fragmented states. In the figure, we observe that the protocol fragments the system in the same parameter regions where circulation is produced. This fragmentation indicates that the system is being excited and, in most cases, the state cannot be described as a product of single-particle states, which is an already expected result. We also observe that for sufficiently small driving frequencies ($\omega_{xy} < 0.1\,J/\hbar$), the system remains condensed, which is a fingerprint of the onset of a critical excitation velocity of superfluid condensates.

To recognize if the optimal realization of the protocol can be well captured by the mean-field model, we have represented in Fig. 11 the final circulation produced in the system with a shadowed area that represents the instances in which the condensed fraction of the system is less than 0.75. These two figures clearly show that in most cases where circulation is produced, the system is fragmented and the mean-field model breaks down. Despite that, the big lobe in which we are interested is not fully covered by the shadowed region, meaning that the approximation can still be useful in parts of the relevant parameter space. Moreover, the results of the circulation production are almost identical for the cases where the system is condensed.

To extract useful information with the mean-field model, one has to restrict the calculations to the region where the system remains condensed. Large differences appear outside this region, where the model is not valid. For example, looking at the best overall configurations, [yellow stars in Fig. 11], we can observe that there is a clear mismatch between the mean-field and exact results. Nevertheless, within its valid regime, the mean-field approach remains a valuable tool for exploring large systems that would otherwise be computationally inaccessible.

SciPost Phys. **19**, 059 (2025)

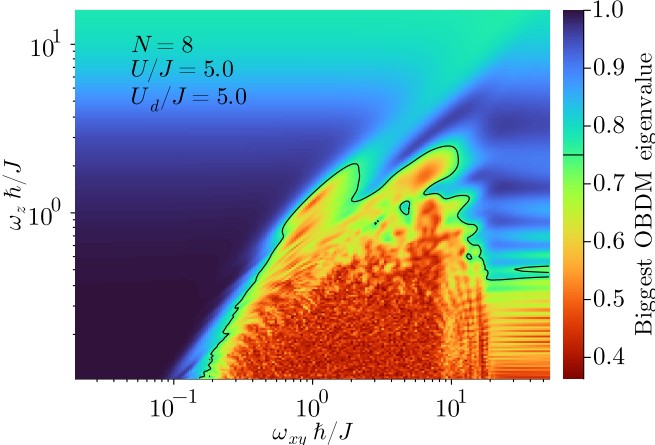

Figure 10: Condensed fraction $f_c$ obtained with exact Bose-Hubbard model at the end of the protocol. The threshold of $f_c = 0.75$ is highlighted with the black line.

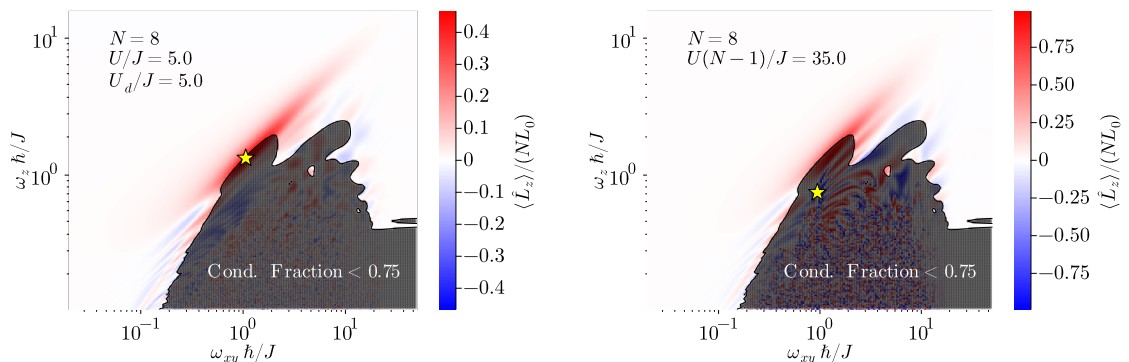

Figure 11: Comparison between the final circulation obtained at the end of the protocol using exact Bose-Hubbard (left panel), and coherent mean-field model (right panel). The shadowed region marks those protocol parameters for which the final condensed fraction is less than 0.75. The yellow star shows the position of the optimal parameters.

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
