# Peer review of "Dipolar magnetostirring protocol for three-well atomtronic circuits"

_SciPost Physics, doi:SciPost Phys. 19, 059 (2025)_

## Round 1 · Referee Report · Anonymous (Referee 1) · 2025-3-10

Report

Dipolar magnetostirring protocol for three-well atomtronic circuits Hector Briongos-Merino, Felipe Isaule, Montserrat Guilleumas, Bruno Juli´a-D´ıaz

The authors develop a novel method to generate persistent currents in a ring-shaped ultracold Bose-Einstein condensate (BEC) using dipolar interactions. Through leveraging the anisotropic, long-range dipole–dipole interactions, the authors propose inducing superfluid flow via rotating the orientation of the dipoles, i.e. “magnetostirring”. The authors explore the proposal numerically in a ring of three potential wells, the minimal system required to have flow, within the extended Bose-Hubbard model and confirm their predictions with a mean-field analysis.

The authors describe a robust protocol by identifying the optimal parameters and conditions for maximal circulation. They develop a method to predict the optimal stirring frequencies, and how to scale for larger systems. This enables applying the scheme to larger number of bosons and more realistic systems.

In my opinion the paper satisfies the criterion to be published in SciPost and would be of interest to a broad range of physicists. The work is interesting, the calculations appear correct and thorough, and in my opinion is an excellent addition to the field of atomtronics and the authors previous work (see Ref[33] for example).

The manuscript is mostly well written, there are some minor typos to be corrected. In my opinion the authors need to be more careful referencing. For example, some additional references for the extended Bose-Hubbard Hamiltonian and the derivation of the transformed Hamiltonian (Eqs. 3 and 4), it would be useful for more citations of the single particle eigenstates, the free boson circulation calculation, and GP equation.

Additionally, the works of: Srivatsa B Prasad, Thomas Bland, Brendan C Mulkerin, Nick G Parker, Andrew M Martin PRA 100 023625 (2019) Srivatsa B Prasad, Brendan C Mulkerin, Andrew M Martin PRA 103 033322 (2021) should be cited when referring to the magnetostiring in dipolar BECs (second page fifth paragraph). In particular, the paper PRA 103 033322 (2021) is cited by Ref. [34] as the motivation of their protocol (along with another PRL by the same authors, however this is not as relevant here.).

Minor comments and questions 1. The authors provide a very broad conclusion on the use of realizing “their utility as rotation or gravity sensors” - this seems a very broad statement. Can the authors expand on how the atomtronic system would be used as a rotation or gravity sensor? Or indeed, the experimental feasibility of their protocol. 2. The figures are mostly clear and well described, however the font seems too small, especially in the insets. The authors should increase the font sizes. 3. In systems with many sites, or a continuous toroidal condensate, do the authors expect their methodology to still produce a persistent current?

Overall, this manuscript represents a relevant contribution, suitable for publication in SciPost after addressing these minor issues.

Recommendation

Publish (easily meets expectations and criteria for this Journal; among top 50%)

  • validity: high
  • significance: high
  • originality: high
  • clarity: high
  • formatting: excellent
  • grammar: excellent

Author:  Héctor Briongos-Merino  on 2025-04-23  [id 5402]

(in reply to Report 1 on 2025-03-10)
Category:
answer to question
correction
suggestion for further work

We thank the Referee for the careful reading of our manuscript and the positive report. We appreciate the effort in highlighting the novelty and contributions of the work, and the recommendation in favor of publication in SciPost Physics. We would also like to express our gratitude for pointing out some references that we missed in our previous version. We have taken into account all the suggestions of the Referee and included extra references to support the extended Bose-Hubbard Hamiltonian, the transformed Hamiltonian, the single particle eigenstates, and the mean-field equation. The font sizes of the figures have also been increased.

In the following lines, we answer the questions:

  • The authors provide a very broad conclusion on the use of realizing “their utility as rotation or gravity sensors” - this seems a very broad statement. Can the authors expand on how the atomtronic system would be used as a rotation or gravity sensor? Or, indeed, the experimental feasibility of their protocol.

The applicability of the persistent currents in atomtronic rings as rotation sensors has been proposed previously in the literature [12, 14]. The utility as a gravity sensor has been proposed for interferometric setups (see Ref. doi: 10.1109/PLANS.2012.6236861). However, we agree that it is a broad statement, and thus, we have removed such a reference. The protocol can be experimentally realized on setups that can create a connected three-well system using, for example, DMDs. The rotation of the dipoles can be achieved by adding a magnetic field, as done with the experimental BEC setup of Prof. F. Ferlaino's group [35]. We have added an explicit mention of this in the conclusions.

  • In systems with many sites, or a continuous toroidal condensate, do the authors expect their methodology to still produce a persistent current?

In systems with a larger number of sites, we expect the protocol to work similarly by properly choosing the initial in-plane polarization direction. We also expect the optimal frequencies to change. Systems with a larger number of sites will be examined in detail in future works. For a continuous toroidal condensate, we also expect the protocol to work for in-plane rotations that are faster than the system's critical rotation frequency. In this system, circulation will appear when one or more vortices are able to cross the toroidal condensate to reach the center of the ring. Therefore, the driving protocol must be fast enough to overcome the energy barrier associated with that vortex trajectory.

Following this question, we have added some comments to the conclusions addressing the feasibility of these larger systems and of toroidal geometries.

---

## Round 1 · Referee Report · Anonymous (Referee 2) · 2025-3-12

Strengths

1 -- All the results are sound and clearly discussed in detail

2 -- The underlying many-body theory is generally explained well.

3 -- The manuscript is well written, including the included appendices for the mean-field description

4 -- Could be experimentally tested within the state-of-the-art could atoms infrastructure.

Weaknesses

1 -- The manuscript is too technical. Whilst the protocol is good, it is also heavily reliant on numerical parameter optimization. The manuscript would greatly benefit if there were analytical results such as for example having a clear microscopic derivation for the parameter prediction to back up the numerics.

2 -- The section of the `Circulation creation protocol' lacks a bit of clarity.

Report

In this work, the authors present a new protocol to generate persistent currents in a ring-shaped lattice by applying magnetostirring. Extending the scheme utilized to generate vortices in diploar gases, the authors demonstrate how a rotation can be induced in the condensate. They perform an extensive investigation of this protocol (robustness and performance) for a wide range of parameters and operating conditions. The results achieved are of great interest to the cold atoms community and present a new direction atomtronics. Given
the strengths and weaknesses outlined above, I think the manuscript as it is would be better suited in SciPost Physics Core.

Recommendation

Accept in alternative Journal (see Report)

  • validity: good
  • significance: high
  • originality: good
  • clarity: good
  • formatting: excellent
  • grammar: excellent

Author:  Héctor Briongos-Merino  on 2025-04-23  [id 5403]

(in reply to Report 2 on 2025-03-12)
Category:
answer to question
reply to objection

We thank the Referee for the careful reading of our manuscript and the constructive criticism of our work. We are pleased that the Referee recognizes the significance of the proposed protocol to generate persistent currents in a ring-shaped lattice via magnetostirring, as well for recognizing that the protocol is experimentally feasible within the state-of-the-art cold atoms infrastructure. We sincerely appreciate the Referee for claiming that the results achieved are of great interest to the cold atoms community and present a new direction in atomtronics, highlighting the synergetic link between these two research areas, which is one of the journal's expectations.

In the following, we address the main points raised:

  • The manuscript is too technical. Whilst the protocol is good, it is also heavily reliant on numerical parameter optimization. The manuscript would greatly benefit if there were analytical results, such as for example having a clear microscopic derivation for the parameter prediction to back up the numerics.

We agree that the protocol involves numerical optimization. We study the system numerically as the extended Bose-Hubbard model is non-integrable for arbitrary dipolar angles. In addition, performing time evolutions in such systems is only feasible with perturbative approaches or other approximations, which are not valid for the strongly-interacting regime studied in the work. We have modified the last paragraph of section 2 to highlight the need to rely on numerics for an accurate examination of the system. Despite the absence of analytical predictions, we still have constraints for the parameters needed to generate circulation based on known behaviors. As a lower bound, if the Hamiltonian changes slowly enough, the system remains in the ground state of the instantaneous Hamiltonian. As an upper bound, at high rotation frequencies, the bosons cannot follow the rapid variation of the polarization direction, causing the system's evolution to resemble that under a static antidipolar interaction. We have rewritten the discussion of the parameters in section 4.2 to clarify the physical regimes involved, focusing on the known limits to bound the optimal parameters.

  • The section of the `Circulation creation protocol' lacks a bit of clarity.

In the revised manuscript, we have modified this section to improve clarity and readability. Our goal is to ensure the protocol is easily understood by a broader audience within the cold atoms community.

We hope that the revisions adequately address the Referee's comments and confirm that the manuscript now meets the standards and expectations of SciPost Physics.

---

## Round 2 · Referee Report · Anonymous (Referee 2) · 2025-6-11

Report

The authors have implemented several changes to their manuscript to improve both the readability and to clarify certain aspects of their work. Specifically, they addressed my comment on their work being too reliant on numerical optimization, introducing it in the last paragraph of Section 2. Furthermore, they revised section 4.2 to highlight the constraints enforced on the numerical optimization for the generation of currents. The authors also revised section 3, giving a better clarification of their protocol and making it easier to understand.

Overall, the revisions to the manuscript implemented by the authors address the comments I raised in the previous round. I think that the current version of the paper meets the criteria and standards to be published in SciPost Physics.

Recommendation

Publish (meets expectations and criteria for this Journal)

---

## Round 2 · Referee Report · Anonymous (Referee 1) · 2025-6-12

Report

The authors have addressed my comments and suggestions. They have also made a clear and visible effort to clarify the article, improving the more technical parts to answer Referee 2's critiques.

On Referee 2’s suggestion to transfer the paper to SciPost Physics Core: I understand the concern that the manuscript is quite technical in parts and relies on numerical optimisation, but I believe the protocol requires numerical simulation and analytical results would be very difficult to derive.

With this in mind, the manuscript is of relevance to the main SciPost Physics audience and can be published.

Recommendation

Publish (easily meets expectations and criteria for this Journal; among top 50%)

---

## Round 2 · List of Changes

• Increased axis label size on all figures.
    • Missing relevant references were added.
    • The last paragraph of section 2 was largely rewritten to better explain the necessity of performing numerical calculations.
    • Several sentences of section 3 were rewritten to improve clarity.
    • Several sentences of section 4.2 were rewritten to better explain the expected values of the protocol parameters.
    • A sentence was added to the conclusions to address the experimental feasibility explicitly.
    • The last paragraph of the conclusions was rewritten to stress extensions of this work.
    • Small changes of style and typos were fixed throughout the text.
    • The appendices were rewritten to improve clarity.
    • The manuscript now uses the official Scipost template.

New references: [12] Pelegrí G, Mompart J and Ahufinger V 2018 New Journal of Physics 20 103001 [36] Prasad S B, Bland T, Mulkerin B C, Parker N G and Martin A M 2019 Physical Review A 100 023625 [37] Prasad S B, Mulkerin B C and Martin A M 2021 Physical Review A 103 033322 [44] Góral K, Santos L and Lewenstein M 2002 Physical Review Letters 88 1 [45] Maik M, Buonsante P, Vezzani A and Zakrzewski J 2011 Physical Review A 84 053615 [46] Lahaye T, Pfau T and Santos L 2010 Physical Review Letters 104 170404 [47] Tonel A P, Ymai L H, Wittmann K, Foerster A and Links J 2020 SciPost Physics Core 2 003 [50] Ferrando A 2005 Physical Review E 72 036612 [55] Paraoanu G S 2003 Physical Review A 67 023607

---

## Editorial Decision

published